# Failure of an Approximately Six Week Course of Tafenoquine to Completely Eradicate *Babesia microti* Infection in an Immunocompromised Patient

**DOI:** 10.3390/pathogens11091051

**Published:** 2022-09-15

**Authors:** Prithiv J. Prasad, Gary P. Wormser

**Affiliations:** 1Division of Infectious Diseases, NYU Grossman School of Medicine, 550 First Avenue, New York, NY 10016, USA; 2Division of Infectious Diseases, New York Medical College, Valhalla, NY 10595, USA

**Keywords:** tafenoquine, babesiosis, *Babesia microti*, neutropenia, leukopenia

## Abstract

Although tafenoquine was highly effective for eliminating microscopically detectable parasitemia in mouse models of *Babesia microti* infection, all of the mice which were assessed developed a relapse of infection, except for those which had been treated concomitantly with artesunate. We report an immunocompromised patient with a similar relapse of parasitemia despite a 46-day course of tafenoquine treatment. More data on whether a longer duration of tafenoquine treatment or using a higher maintenance dose, versus adding a second drug to the regimen, will prevent relapse when tafenoquine is used to treat a highly immunocompromised patient with babesiosis should be investigated.

## 1. Introduction

Even just one dose of the United States Food and Drug Administration approved drug tafenoquine is highly effective in rapidly clearing *Babesia microti* parasites from the blood of infected mice [1]. However, in mouse studies in which from one to three doses of tafenoquine were administered, the parasite was not eliminated, with the potential for parasitemia to recur [1,2]. Whether longer courses of tafenoquine would completely eradicate viable *B. microti* parasites has not been studied in murine models.

Here we report on an immunocompromised patient with multiple relapses of *B. microti* infection despite the use of combination therapy with medications recommended to treat *B. microti* infections [3]. The patient was then treated with tafenoquine as a single drug with the intention to treat for at least three months. However, only approximately six weeks of treatment was possible due to the development of neutropenia. Despite the resolution of all laboratory abnormalities indicative of active babesiosis, and despite three negative polymerase chain reaction (PCR) tests over a 14-day interval, relapse of the *B. microti* infection occurred based on both PCR and blood smear testing.

## 2. Case Summary

A 74-year-old woman was hospitalized on 29 August 2021 because of fatigue and fever for two days. She had a history of diffuse large B-cell lymphoma treated with chemotherapy (six cycles of R-CHOP regimen (rituximab plus cyclophosphamide, doxorubicin hydrochloride, vincristine sulfate, and prednisone) with her last cycle on 12 March 2018), polymyalgia rheumatica treated with low dose steroids (prednisone 5 mg/day, plus a short trial of tocilizumab from 5 October 2020 to 3 December 2020), cold autoimmune hemolytic anemia diagnosed on 13 July 2021 when she was hospitalized and evaluated for fever and severe hemolytic anemia. She received multiple blood transfusions (10U PRBC in total, last on 27 July 2021) and ultimately stabilized with prednisone 60 mg/day, which had been tapered to 5 mg/day by the time of the current hospitalization.

During the current hospitalization, the initial hemoglobin level was 6.5 g/dL. A peripheral blood smear was positive for *B. microti* (0.2% parasitemia). Since she was not considered to be significantly immunocompromised at that time due to her low-dose steroid use, she completed one week of azithromycin and atovaquone from 1 September 2021 to 7 September 2021 (Table 1). She became afebrile after starting antiparasitic drug therapy; a peripheral blood smear was negative for *B. microti* on 4 September 2021. Her prednisone dose was increased to 20 mg/day during her hospitalization due to increased hemolysis, and the hemoglobin level was stable at 8.6 g/dL at discharge. She continued tapering steroids on discharge. IgG antibodies to *B. microti* were repeatedly detected starting on 3 September 2021, suggesting that the immunosuppressive effects of the rituximab had waned. How she acquired babesiosis is unclear since she had no exposure to an *Ixodes scapularis* tick endemic area for the preceding five years, raising the question of whether she was infected through a blood transfusion.

She developed recurrent fatigue and fever (100.2 F) on 10 September 2021, and a peripheral blood smear showed a recurrence of parasitemia (0.3%). She resumed azithromycin and atovaquone on 11 September 2021 (Table 1). She continued on azithromycin and atovaquone for the planned six-week course; however, parasitemia persisted along with severe fatigue. Therefore, on 11 December 2021 oral clindamycin was added and continued for another week. Nevertheless, parasitemia persisted, and therefore her antiparasitic regimen was switched to quinine plus oral clindamycin (Table 1). Peripheral blood smears became negative by 28 December 2021. This antiparasitic drug regimen was discontinued on 20 January 2022, because she developed symptoms consistent with cinchonism (hearing loss, vertigo, tinnitus). She had had four weeks of negative peripheral blood smears at this point, and had a normal hemoglobin level. She felt well symptomatically, and her prednisone dose was tapered and stopped on 5 February 2022. Symptoms of cinchonism resolved completely after discontinuing quinine.

She then developed severe fatigue and subjective fevers on 22 February 2022 with a recurrence of the babesia parasitemia (<0.1%), along with evidence of worsening hemolysis. On 24 February 2022, she was restarted on oral clindamycin, 300 mg orally every 8 h, along with a reduced dose of quinine of 324 mg every 8 h. However, peripheral blood smears remained positive for *B. microti* (parasitemia < 0.1%) on 3 March 2022. A babesia PCR assay was obtained for the first time (performed by a commercial laboratory), which was also positive. A shared decision was made to attempt off-label use of tafenoquine, since she had tested negative for glucose-6-phospate dehydrogenase deficiency. She was not on any immunosuppressive medication at this time point, with the last dose of prednisone received on 5 February 2022. She was started on oral tafenoquine 200 mg once a day for 3 days (loading dose) from 7–9 March 2022, and then 200 mg once per week thereafter starting on 16 March 2022. Her fatigue improved by the end of week one after starting tafenoquine, and the peripheral blood smear was negative for parasites on 10 March 2022. Babesia PCR became negative by 6 April 2022 and remained negative on subsequent weekly samples over the following two-week period. When tafenoquine was initiated, the hemoglobin level was 6.3 g/dL, which increased to 13.3 g/dL by 14 April 2022 without blood transfusions. The lactate dehydrogenase level (LDH) had also normalized.

However, she developed neutropenia (neutrophil cell count of 641 cells/uL with a total white blood cell count of 1.7 × 10^3^/uL) on 20 April 2022, although without any new clinical symptoms. Tafenoquine was discontinued on 21 April 2022, and the neutrophil count reached a nadir level of 325 cells/uL on 27 April 2022, but recovered to 1300 cells/uL by 4 May 2022. The original plan had been to treat with tafenoquine for at least three months.

She continued to have weekly monitoring blood draws and was noted to have a positive *B. microti* PCR test result on 11 May 2022 (Ct 34.2), with Ct values declining over the next two weeks (Ct 29.9 on 19 May 2022 and 27.0 on 25 May 2022). She had early signs of worsening hemolysis (haptoglobin level of <8 mg/dL) on 25 May 2022 but remained asymptomatic at this time. By 1 June 2022, however, she developed recurrent severe fatigue and had further evidence of hemolysis (LDH rose to 1.5 times the upper limit of normal). She was started on a new drug regimen of oral azithromycin 1000 mg once daily, atovaquone liquid suspension 750 mg once daily, and four Malarone^®^ tablets once daily (each tablet consisting of 250 mg atovaquone plus 100 mg of proguanil) on 2 June 2022. A peripheral blood smear was positive for *B. microti* (<0.1%) on 7 June 2022; however, the patient reported a marked improvement in fatigue after restarting antiparasitic therapy, which continues to date. Babesia PCR testing was negative on 22 June 2022, and remains negative to date (as of the last blood draw on 8 September 2022). It was planned to continue treatment for at least 12 weeks and even to consider chronic suppressive therapy going forward.

## 3. Discussion

Besides the patient described in this case study, only two other patients with chronic relapsing *B. microti* infection who were treated with tafenoquine have been reported [4,5]. Both were regarded as having a successful outcome, but limitations existed for both cases precluding a definitive conclusion as to the efficacy of tafenoquine per se. Both patients had clinical and molecular evidence of resistance to certain of the first-line drugs recommended to treat *B. microti* infections. The first reported case had a prior history of vasculitis and had received two doses of rituximab, the last of which was administered in January 2017, two years before he was diagnosed with babesiosis [4]. Rituximab therapy has been associated with chronic relapsing babesiosis lasting for up to two years or longer [6]. In late January 2020, because of multiple relapses, he was begun on treatment with a six-week course of a novel four-drug regimen that included Malarone^®^ plus azithromycin (1000 mg orally per day) plus clindamycin (450 mg orally three times per day) plus 750 mg of atovaquone per day (in addition to the 1000 mg of atovaquone included the daily dosage of Malarone^®^). The patient had seroconverted for antibodies to *B. microti* sometime between 11/10/19 and 1/28/20, suggesting that the humoral immunosuppression from rituximab had waned. The patient developed nausea and diarrhea by day 41 of this regimen, and then was changed to tafenoquine alone, which was administered for six more weeks starting on 3/10/20 (200 mg daily for three days followed by 200 mg weekly). PCR testing on 2/28/20 was positive but was repeatedly negative during the tafenoquine treatment period and remained negative approximately one month after completion of the tafenoquine regimen. The patient remained well over nearly 19 months of follow-up after completion of tafenoquine. Because of the improvement the patient experienced on the Malarone^®^-based drug regimen, along with evidence for reduced immunosuppression from the rituximab, the authors emphasized that it was impossible to assess the impact the tafenoquine might have had on the successful outcome [4].

The second patient had been treated with immunosuppressive therapy that included rituximab in late 2020 for monoclonal B-cell lymphocytosis found on a bone marrow biopsy specimen [5]. Five months later, in May 2021, the patient was diagnosed with babesiosis and was treated unsuccessfully on multiple occasions using various recommended drug regimens [3]. On day 256 after diagnosis of the *B. microti* infection, tafenoquine was added to other anti-babesia drugs, but 13 days later, tafenoquine was given as a single agent. The total course of tafenoquine given was approximately nine weeks. The patient remained well thereafter, but there was only a four-week follow-up period following the completion of treatment. In addition, babesia antibody testing was not reported precluding an assessment of whether the humoral immunosuppressive effects of rituximab had waned. The tafenoquine dosage regimen this patient received included a 600 mg loading dose given over three days followed by 300 mg per week versus 200 mg per week used in the first reported case [4] and also used for our patient. Understanding whether the 50% higher maintenance dosing may have had a favorable impact on the longer-term outcome is of interest.

The patient described in the current report received a 46-day course of tafenoquine, which was discontinued because of the development of neutropenia, which reversed following discontinuation of tafenoquine, suggesting that it was an unexpected adverse reaction from the drug [7]. PCR testing to detect *B. microti* in blood remained negative six days after discontinuing tafenoquine, but became positive by day 21 after discontinuation of tafenoquine.

Therefore, based on available data, tafenoquine as a single agent may, or may not, be curative of *B. microti* infection in a chronically immunocompromised patient. In a non-immunocompromised mouse model, *B. microti* infection appeared to be fully eradicated when a three-day tafenoquine regimen was combined with a five-day course of artesunate [2]. Clearly, more clinical studies and more studies conducted in animal models are needed to optimize the use of tafenoquine in order to prevent a relapse of *B. microti* infection in chronically immunocompromised patients with babesiosis when the drug is discontinued.

## Figures and Tables

**Table 1 pathogens-11-01051-t001:** Summary of babesiosis treatment courses (all oral).

Initiation Date and Duration	Reason for Treating with Antiparasitic Drug Therapy	Azithromycin (Dose)	Atovaquone (Dose) *	Clindamycin (Dose)	Quinine	Malarone (Dose) **	Tafenoquine (Dose)
9/1/21 for 7 days	Fever and +blood smear 0.2%	500 mg once per day	750 mg twice per day				
9/11/21 for 30 days	Recurrence of fever and +blood smear 0.3%	500 mg once per day	750 mg twice per day				
12/11/21 for 7 days	Persistent parasitemia	500 mg once per day	750 mg twice per day	300 mg three times per day			
12/19/21 for 33 days; stopped due to cinchonism; smears were negative for 4 weeks	Persistent parasitemia			300 mg three times per day	650 mg three times per day		
2/24/22 for 11 days	Recurrent parasitemia < 0.1%			300 mg three times per day	324 mg three times per day		
3/7/22 for 46 days; stopped due to neutropenia	Persistent parasitemia						200 mg daily for 3 days, then 200 mg once per week
6/2/22 for 26 days	Recurrent PCR positivity, rising LDH, and fatigue. + blood smear on 6/7/22 < 0.1%	1000 mg once per day	750 mg once per day			4 pills daily	
6/27/22 and ongoing	Change in dosage due to GI symptoms	500 mg once per day ***	750 mg once per day			4 pills daily	

* Liquid preparation; ** Each Malarone^®^ tablet contains 100 mg of proguanil plus 250 mg of atovaquone; *** Azithromycin was discontinued on 7/27/22; LDH = Lactate dehydrogenase; PCR = Polymerase chain reaction to detect *Babesia microti;* GI = Gastrointestinal.

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
