# Peer review of "Failure of an Approximately Six Week Course of Tafenoquine to Completely Eradicate Babesia microti Infection in an Immunocompromised Patient"

_pathogens, 2022, doi:10.3390/pathogens11091051_

Round 1

Reviewer 1 Report

The author report on an immunocompromised patient was treated with tafenoquine as a single drug and failed to completely eradicate Babesia microti infection. I have few comments may help to improve the manuscript.

1: Please discuss why neutropenia developed with tafenoquine or other drugs treatment.

2: Did the patient tested G6PD deficiency, if so, please add the information in main text.

3: Recurrence is much better than relapse for Babesia, relapse is used for malaria.

4: Single drug of Tafenoquine failed to completely eradicate Babesia rodhaini and Babesia gibsoni infections in immunocompromised hosts were reported. So, I think it would be similar for B. microti infection in immunocompromised patient.

Author Response

The author report on an immunocompromised patient was treated with tafenoquine as a single drug and failed to completely eradicate Babesia microti infection. I have few comments may help to improve the manuscript.

1: Please discuss why neutropenia developed with tafenoquine or other drugs treatment.

RESPONSE: This topic is a bit too complicated for this case report

2: Did the patient tested G6PD deficiency, if so, please add the information in main text.

RESPONSE: This information is now provided.

3: Recurrence is much better than relapse for Babesia, relapse is used for malaria.

RESPONSE: The word “relapse” is most often used for babesiosis cases, see the IDSA Babesiosis Guideline and also this reference: Persistent and relapsing babesiosis in immunocompromised patients; Clin Infect Dis 2008;46:370-6.

4: Single drug of Tafenoquine failed to completely eradicate Babesia rodhaini and Babesia gibsoni infections in immunocompromised hosts were reported. So, I think it would be similar for B. microti infection in immunocompromised patient. 

RESPONSE: Specifically relapse has occurred with Babesia microti in murine studies, as discussed in the manuscript.

Reviewer 2 Report

The manuscript presented by Prasad and Wormser refers to a case study in which babesiosis treatment, in an elderly woman, with different therapeutic schemes have failed. The case is well described including all relevant clinical information which in turn supports the treatment choices. Treatment of babesiosis in immunocompromised patients is recognized as difficult, with high risk of persistent relapsing illness. Resistance prediction to azithromycin, clindamycin and atovaquone have been recently which may and should support therapeutic choices in these more complex cases. In fact, seems that resistance is on the rise and with the increasing numbers of human babesiosis (I believe that particularly in North America) this is a very important issue that needs to be taken in account. Thus, it would have been very important if this analysis had been done. Even though authors refer prior studies in which resistance was evaluated authors can highlight the importance of this and discuss it.

The particularity of the cases in which the drug tafenoquine was used does not allow concluding about its effectiveness but shows the need to carry out studies that clarify about dosages, possible combine therapy among others. My only concern with the ms is that therapy is still in course, and it would be more informative and final if authors delay the ms to include this data. As authors realize for future reference this is of the outmost importance, otherwise doubt is always to be present.  

Hope that the patient is doing well.   

Author Response

The manuscript presented by Prasad and Wormser refers to a case study in which babesiosis treatment, in an elderly woman, with different therapeutic schemes have failed. The case is well described including all relevant clinical information which in turn supports the treatment choices. Treatment of babesiosis in immunocompromised patients is recognized as difficult, with high risk of persistent relapsing illness. Resistance prediction to azithromycin, clindamycin and atovaquone have been recently which may and should support therapeutic choices in these more complex cases. In fact, seems that resistance is on the rise and with the increasing numbers of human babesiosis (I believe that particularly in North America) this is a very important issue that needs to be taken in account. Thus, it would have been very important if this analysis had been done. Even though authors refer prior studies in which resistance was evaluated authors can highlight the importance of this and discuss it.

RESPONSE: Unfortunately, we did not perform resistance testing in this case.

The particularity of the cases in which the drug tafenoquine was used does not allow concluding about its effectiveness but shows the need to carry out studies that clarify about dosages, possible combine therapy among others. My only concern with the ms is that therapy is still in course, and it would be more informative and final if authors delay the ms to include this data. As authors realize for future reference this is of the outmost importance, otherwise doubt is always to be present.  

RESPONSE: We did update the last PCR testing result.  

Hope that the patient is doing well.   

RESPONSE:  Doing very well.